# Quantifying the Relationship between SARS-CoV-2 Wastewater Concentrations and Building-Level COVID-19 Prevalence at an Isolation Residence: A Passive Sampling Approach

**DOI:** 10.3390/ijerph191811245

**Published:** 2022-09-07

**Authors:** Patrick T. Acer, Lauren M. Kelly, Andrew A. Lover, Caitlyn S. Butler

**Affiliations:** 1Department of Biostatistics and Epidemiology, University of Massachusetts Amherst, Arnold House, 715 North Pleasant Street, Amherst, MA 01003, USA; 2Department of Environmental and Water Resources Engineering, University of Massachusetts Amherst, Engineering Lab II, 101 N Service Rd, Amherst, MA 01003, USA

**Keywords:** SARS-CoV-2, college campus monitoring, passive sampling, wastewater-based epidemiology, fecal shedding, COVID-19

## Abstract

SARS-CoV-2 RNA loads can be detected in the excreta of individuals with COVID-19 and have demonstrated positive correlations with clinical infection trends. Consequently, wastewater-based epidemiology (WBE) approaches have been implemented globally as a public health surveillance tool to monitor community-level prevalence of infections. The majority of wastewater specimens are gathered as either composite samples via automatic samplers (autosamplers) or grab samples. However, autosamplers are expensive and can be challenging to maintain in cold weather, while grab samples are particularly susceptible to temporal variation when sampling sewage directly from complex matrices outside residential buildings. Passive sampling can provide an affordable, practical, and scalable sampling system while maintaining a reproducible SARS-CoV-2 signal. In this regard, we deployed tampons as passive samplers outside of a COVID-19 isolation unit (a segregated residence hall) at a university campus from 1 February 2021–21 May 2021. Samples (n = 64) were collected 3–5 times weekly and remained within the sewer for a median duration of 24 h. SARS-CoV-2 RNA was quantified using reverse-transcription quantitative polymerase chain reaction (RT-qPCR) targeting the N1 and N2 gene fragments. We quantified the mean viral load captured per individual and the association between the daily viral load and total persons, adjusting for covariates using multivariable models to provide a baseline estimate of viral shedding. Samples were processed through two distinct laboratory pipelines on campus, yielding highly correlated N2 concentrations. Data obtained here highlight the success of passive sampling utilizing tampons to capture SARS-CoV-2 in wastewater coming from a COVID-19 isolation residence, indicating that this method can help inform building-level public health responses.

## 1. Introduction

Severe acute respiratory syndrome coronavirus 2 (SARS-CoV-2) is the seventh coronavirus known to infect human beings and the causative agent for the COVID-19 pandemic [1,2]. While COVID-19 is most notably known for causing respiratory illness, more recently, it is being recognized as a multi-organ disease with a broad range of extrapulmonary manifestations, including gastrointestinal symptoms [3]. The pooled prevalence of gastrointestinal symptoms in COVID-19 patients is about 15%, with diarrhea, loss of appetite, and nausea/vomiting being the most common symptoms. The mean duration of SARS-CoV-2 RNA shedding in stool is estimated to be roughly 17 days; however, as new variants continue to emerge, shedding durations will fluctuate [4]. However, prolonged fecal shedding of SARS-CoV-2 RNA has been documented up to 47 days following symptom onset, sometimes persisting in stool for more than two weeks after negative oropharyngeal swabs [2,4,5,6,7,8]. It has been estimated that SARS-CoV-2 RNA in stool is detected in nearly 66% of infected patients and can be measured in feces as soon as two days after symptom onset [9,10,11]. Evidence suggests that both symptomatic and asymptomatic individuals shed SARS-CoV-2 in both respiratory and stool specimens, highlighting the critical role wastewater-based epidemiology (WBE) can play in capturing asymptomatic or subclinical cases that may otherwise not be identified [8,9,12,13,14,15,16]. Yet, differences in viral shedding rates based on symptom status and disease time course have been detailed, making it difficult to quantitatively interpret SARS-CoV-2 wastewater concentrations, especially to estimate infection prevalence [17]. Moreover, additional predictor variables capturing community infection dynamics, disease progression, temporal variations, and fecal-shedding kinetics are likely important to consider, if possible, when estimating caseloads based on SARS-CoV-2 wastewater signatures [18,19,20,21].

Many higher educational institutions and municipalities globally have recognized the utility of WBE as a surveillance tool for monitoring infection trends and identifying potential case clusters promptly. Several studies have demonstrated a positive correlation between SARS-CoV-2 RNA concentrations in influent wastewater/settled solids and COVID-19 case metrics, demonstrating the utility of this approach in complementing clinical reporting, particularly in settings where distinct buildings can be isolated [22,23,24]. Additionally, in the absence of widespread clinical testing, WBE can act as a health surveillance tool to monitor viral dynamics, as it has been shown to presage new case and subsequent hospitalization rates [22,25]. Accordingly, numerous higher educational institutions have adopted building-level wastewater testing in combination with contact tracing and isolation/quarantine protocols to support campus public health and guide clinical testing resources to “hotspot” locations [25,26,27,28,29,30]. Due to its ability to capture population-level health information, WBE will continue to serve as a valuable means to monitor and inform public health programming and response.

WBE is ideally suited to complement widespread clinical testing campaigns. Sample collection strategies vary with three predominant methodologies: composite sampling, grab sampling and passive sampling. Autosamplers can mechanically pull wastewater samples routinely with defined time or flow intervals to create representative composite aliquots; however, these instruments can be expensive and challenging to maintain in certain environments [31]. Grab samples are generally an operationally straightforward option, collecting a volume of wastewater at a single timepoint. However, such samples are susceptible to fecal shedding and flow fluctuations, particularly outside buildings with limited drainage areas that may provide a biased sample of the wastewater stream [31,32]. Several universities have trialed the use of passive samplers in place of traditional composite or grab sampling methods due to additional cost-effectiveness and ease of deployment [31,33,34,35,36]. Schang et al. demonstrated that the concentration of SARS-CoV-2 RNA recovered from passive samplers was positively correlated with the viral wastewater concentrations obtained using conventional sampling methods in communities with low caseloads [37]. Habtewold et al. demonstrated linear uptake of SARS-CoV-2 virus utilizing gauze and membrane passive devices when deployed between 4 and 48 h, indicating that passive sampling methods can accumulate SARS-CoV-2 viral fragments over time [38]. In addition, Rafiee et al. demonstrated that Moore swabs (gauze pads with string) performed as well as composite samples over 16 h and were more sensitive than grab samples [32]. A passive sampling approach may be the only practical option for capturing a representative wastewater sample in resource-constrained settings. However, questions still remain on the viral loading dynamics (sorption-desorption rates) with such sampling methods. Rigorous evaluation of the relationship between passive samples and epidemiological metrics is needed to ensure optimal use of limited resources.

To address this gap, we provide the results of 64 independent wastewater samples obtained between 1 February 2021–21 May 2021, from a COVID-19 isolation residence. All samples were collected through a passive sampling method using tampons that were placed in the sewer for at least 24 h. The primary aims of this study were to: (i) demonstrate that a passive sampling approach yields consistent positive SARS-CoV-2 signals coming from building-level wastewater monitoring with a known number of infected persons, (ii) describe the variability in individual fecal shedding rates quantified through this sampling method, (iii) investigate the associations between the SARS-CoV-2 RNA concentrations extracted from the tampons and the corresponding caseload in the building, (iv) compare two independent methods for concentrating wastewater samples and subsequent analysis.

## 2. Materials and Methods

### 2.1. Study Site

During the course of this study, wastewater samples were collected at an access manhole location roughly 190 feet from a COVID-19 isolation residence. The isolation residence served as a temporary living space for students who tested positive for COVID-19. On-campus students relocated to this space after receiving positive test results and were instructed to remain in isolation for 14 days. The wastewater influent at this sampling location was restricted to the isolation building such that the only inputs in the sewer system came from either infected individuals residing within the building or potentially a small number of staff supporting the students in isolation. Sample collection occurred throughout the Spring 2021 academic semester. Nearly 90% of samples were deployed for roughly 24 h; however, a minority were deployed for extended periods (weekend). Three to five sampling rounds occurred weekly from 1 February 2021–21 May 2021. Immediately following collection, all samples were stored at +4 °C for 1–3 h until further processing.

### 2.2. Passive Samples and Wastewater Processing

We utilized tampons made from rayon with a polyester string (OB Applicator Free Tampons, Ultra Absorbency). Two tampons were tied onto separate lengths of 1/8-inch nylon paracord and deployed concurrently at the collection site. After roughly 24 h, both tampons were collected and transferred to sterile 500 mL high-density polyethylene (HDPE) wide-mouth bottles (ThermoFisher Scientific, Waltham, MA, USA). Both tampons were mixed with a total of 200 mL of Milli-Q water, vortexed for 1 min at 3200 RPM, and pressed utilizing a citrus squeezer to remove maximum liquid from the tampons. The volume obtained from each sampling event ranged from 52 mL to 80 mL, resulting in a total sample volume ranging from 252 mL to 280 mL after the addition of Milli-Q water. Bovine Respiratory Syncytial Virus (BRSV) was spiked into each sample in a 1:10,000 volume ratio resulting in 1000 BRSV gene copies per mL in the wastewater sample matrix (Inforce 3 Cattle Vaccine, Zoetis Inc., Parsippany-Troy Hills, NJ, USA). BRSV was introduced as a surrogate for SARS-CoV-2, functioning as a matrix recovery control to estimate SARS-CoV-2 losses during processing as well as during sample freeze–thaw cycles. Following the BRSV spike-in, samples were homogenized using a vortex mixer (3200 RPM) and frozen at −80 °C for future processing.

### 2.3. Protocol 1: SARS-CoV-2 Concentration and Quantification

In June 2021, all samples were removed from −80 °C and thawed at +4 °C prior to processing. Once thawed, samples were centrifuged at 5500 RPM in 250 mL HDPE bottles for 10 min at +4 °C. The sample supernatant was decanted into sterile 250 mL HDPE bottles for filtration; the remaining pellet was discarded. We utilized a vacuum filtration assembly and generally followed the concentration procedure outlined in detail by Bivins et al. in their protocol [39]. Briefly, we used one vacuum filtration assembly and filtered 50 mL of the sample through negatively charged 0.45 μM mixed-cellulose ester membrane filters in duplicate (Pall Corporation, Port Washington, NY, USA). Following filtration, the filter was halved and transferred aseptically into a 0.70 mm garnet bead tube in preparation for RNA extraction (Qiagen, Germantown, MD, USA).

Prior to extraction, 500 μL of Buffer AVL (Qiagen, Germantown, MD, USA) and 6.5 μL of β-mercaptoethanol (MP Biomedicals, Irvine, CA, USA) were added to the garnet bead tubes. Tubes were bead beat for 60 s at 3200 RPM for four cycles and centrifuged for 30 s between each homogenization cycle at 5000× *g*. After completion of bead beating, tubes were centrifuged for a final time at 16,000× *g* for 3 min, and 140 μL of sample was transferred into a 1.5 mL microcentrifuge tube in preparation for extraction. RNA was then extracted using Qiagen’s protocol utilizing the QIAamp Viral RNA Mini Kit (Qiagen, Germantown, MD, USA). Purified RNA concentrate was eluted with 80 μL of Buffer AVE (Qiagen, Germantown, MD, USA) and held at +4 °C briefly. Total RNA in each sample was measured using an RNA high sensitivity assay kit on the Qubit 4.0 prior to RT-qPCR quantification (ThermoFisher Scientific, Waltham, MA, USA).

SARS-CoV-2 was quantified using real-time qPCR, applying SYBR chemistry (Luna^®^ Universal One-Step qPCR Master Mix, Cat No. E3005E). The N1 and N2 genes unique to SARS-CoV-2 were quantified using the 2019-nCoV N1 forward (Cat No. 10006830) and reverse (Cat No. 10006831) primers and the 2019-nCoV N2 forward (Cat No. 10006833) and reverse (Cat No. 10006824) primers synthesized by Integrated DNA Technologies (IDT) (Appendix A). Standard curves for the N1 and N2 analyses were generated by quantifying a synthesized SARS-CoV-2 plasmid manufactured by IDT (Cat No. 10006625). The standard was assayed in triplicate following a 10-fold serial dilution from 2000 gene copies/μL to 0.2 gene copies/μL. All RNA samples were assayed in triplicate. Samples were processed in replicate and assayed in triplicate, generating six RT-qPCR data points per individual sample. Each 20 μL reaction mix contained the following: 4 μL of template RNA; 10 μL of 2× Luna Universal One-Step Reaction Mix; 1 μL of 20X Luna WarmStart^®^ RT Enzyme Mix; 1.6 μL of primers; and 3.4 μL of PCR-grade water. PCR analysis was conducted using a BioRad 96-well real-time PCR system, with the following cycle parameters: 55 °C for 10 min; 95 °C for 1 min; 40 cycles × (95 °C for 10 s, 62 °C for 30 s); 60–95 °C in 5 s increasing increments of 0.5 °C. Melt curves were analyzed to ensure the amplification of a single target PCR amplicon. No-template controls (NTCs) were assayed in triplicate for all PCR runs for quality control. Following successful sample quantification, all wastewater samples were again stored at −80 °C. Standard curves were used to quantify N1 and N2 gene copies in the polymerase chain reaction, which were converted to gene copies/L of raw wastewater captured by our passive samplers (Appendix A). All samples that failed to meet the following criteria were re-assayed: (i) standard curves with R^2^ > 0.985, (ii) primer efficiency between 90–120%, (iii) no signs of PCR inhibition or non-specific amplification. Inhibition was evaluated through melt curve analysis (MCA). The y-intercept value for all runs ranged from (35.1, 36.6), and the slope values ranged from (−3.33, −2.93). At 95% confidence, the level of detection (LOD) for this assay was estimated to be $1.6\times 10^{4}$ SARS-CoV-2 gene copies/L of wastewater fitting our data the following sigmoidal function [40]:$$y_{i}=\frac{1}{1+e^{-\infty -\beta \log(c_{i})}}$$

BRSV quantification occurred using an identical RT-qPCR procedure but with custom BRSV primers (Appendix A) (ThermoFisher Scientific, Waltham, MA, USA).

### 2.4. Protocol 2: SARS-CoV-2 Concentration and Quantification

In October 2021, all 64 raw wastewater samples were removed from −80 °C and thawed at +4 °C. These samples were quantified for a second time using a different concentration method through a separate RNA extraction and RT-qPCR pipeline. Once raw wastewater samples were thawed, 40 mL of sample was added to a 50 mL conical tube. 600 μL of affinity-capture magnetic hydrogel nanoparticles in solution at a 5 mg/mL concentration was spiked into the sample (Ceres Nanosciences, Manassas, VA, USA). Samples were homogenized and incubated at RT for 20 min. After incubation, samples were placed on custom-built magnetic racks for 30 min at RT for viral concentration. Magnetic separation allowed for the easy removal of supernatant while retaining a pellet of nanoparticles with the entrapped viral fragments. 1.2 mL of DNA/RNA Shield (Zymo Research, Irvine, CA, USA) was added to each pellet. Samples were briefly homogenized and incubated at 56 °C (ten minutes) to release the viral fragments from the nanoparticle matrices. Samples were returned to the magnetic racks for 10 min to separate the sample lysate from the nanoparticles. 500 μL of the sample was aliquoted into technical replicate test tubes to be further processed at the Institute for Applied Life Sciences Clinical Testing Center (ICTC), which obtained its state clinical laboratory license (CLIA) in October 2020. Briefly, ICTC utilized an automated RNA extraction platform (Hamilton Company, Reno, NV, USA) using the MagMax Viral/Pathogen II Nucleic Acid Isolation Kit (ThermoFisher Scientific, Waltham, MA, USA) to isolate and purify nucleic acid. SARS-CoV-2 was quantified targeting the N2 gene with real-time qPCR, applying TaqMan chemistry (Luna^®^ Universal Probe One-Step RT-qPCR Kit, E3006E) using a BioRad 384-well real-time PCR system. Data were reported as extracted gene copies/mL from our concentrated samples, which we converted to gene copies/L of wastewater by using the known dilution factor from the original sample.

### 2.5. Isolation Residence Case Data and Clinical Surveillance

Aggregated and de-identified COVID-19 case data were obtained from administrative records for the isolation residence on campus during the Spring 2021 academic year. These data consisted of the total number of individuals present in the isolation facility, including sex and self-reported symptoms. The public health program was approved by the Institutional Review Board (#20-258); this research protocol had a separate filing (approval #21-140). Data were processed by HIPAA-trained staff at the Public Health Promotion Center (PHPC) and were acquired on university-administered systems and HIPAA-compliant platforms. During the timeframe of wastewater testing, all on-campus students were required to be tested twice weekly via nasal swab PCR tests. All off-campus students, faculty, and staff that came to campus were required to be tested once weekly during the semester. On-campus students with a positive SARS-CoV-2 test were required to isolate in the designated isolation residence hall. Off-campus students were provided the option to isolate themselves on campus.

### 2.6. Data Analysis

Statistical analyses were performed using SAS 9.4 (SAS Institute, Cary, NC, USA), and visualizations were created using RStudio (ver. 1.4.1103) with ggplot2 (ver. 3.3.5). Slopes and y-intercepts from RT-qPCR were used to quantify copies of SARS-CoV-2 in each reaction using the instrument’s recorded Cq value. The SARS-CoV-2 copy number value was then transformed into a gene copies/L of extracted wastewater value through a series of unit conversations and dilution factors based on our wastewater processing methods (Appendix A). Building-level volumetric water metering provided 24 h building-level water use totals for the isolation residence, which were utilized as a proxy for daily wastewater flow. The flow values were converted from cubic feet to liters. Daily viral loads at the isolation residence were estimated by multiplying [(SARS-CoV-2 gene copies/L of wastewater) × (L of wastewater influent/day)] to provide an aggregate SARS-CoV-2 gene copies/day measurement. Wilcoxon-Mann–Whitney (WMW) tests were utilized to test for statistical differences between the N1 and N2 gene copy concentrations due to the paired comparisons and non-normal distribution of these data. This test was also employed to assess statistical differences between N2 gene concentrations quantified through two separate sample processing and analysis pipelines. We employed multivariable negative binomial models to quantify the relationship between the wastewater viral load and the accompanying building caseload, adjusting for significant covariates. We evaluated the performance of the negative binomial model utilizing McFadden’s pseudo-R^2^ statistic. All tests were two-sided, with α = 0.05 for hypothesis testing.

## 3. Results and Discussion

### 3.1. Student Characteristics & Performance of Passive Samples

During the Spring 2021 academic term, individuals in the isolation residence ranged from 17–25 years of age with females making up 40.6% of total cases and males making up 59.4% of cases on wastewater collection days (Appendix A). During this time, 92.3% of COVID-19 cases reported having at least one illness-related symptom. A global meta-analysis including nearly 30 million individuals undergoing COVID-19 testing found that 40.5% (95% CI, 33.50–47.50%) of laboratory-confirmed COVID-19 cases were asymptomatic [41]. In this meta-analysis, patient symptom reports occurred at multiple timepoints along the care cascade possibly resulting in the misclassification of cases captured prior to symptom onset. In contrast, cases in our isolation residence were asked to report symptoms daily throughout their illness, which may partially explain the discordance between the high proportion of symptomatic individuals in the isolation residence compared with estimates of symptomatic prevalence found in the general population.

Overall, the tampon swabs consistently captured the SARS-CoV-2 molecular signature from the waste stream. Each sample contained liquid from two tampons yielding between 52 mL and 80 mL of raw sewage. Total RNA mass extracted from each sample ranged from 1.31 ng/μL to 35.5 ng/μL with a mean concentration of 7.0 ng/μL ± 5.14, indicating that the tampons captured fluctuating amounts of RNA in the sewer. Over the 16-week study period, all passive samples captured SARS-CoV-2, demonstrating consistency in tampons amassing detectable amounts of virus. The mean N1 Cq value was 32.84 ± 2.35, and the mean N2 Cq value was 32.37 ± 2.57 for all datapoints obtained using Protocol 1. A Wilcoxon rank-sum test indicated no evidence of a statistically significant difference between the median N1 and N2 signals within this study (*p* = 0.69). Additionally, the mean N2 Cq value provided by ICTC on identical raw wastewater samples processed through Protocol 2 was 26.53 ± 2.63. The increased N2 Cq sensitivity observed using Protocol 2 is due to sample processing differences between the two methods. However, as detailed later, generally the final SARS-CoV-2 quantification between both assays is similar. Overall, the variability of SARS-CoV-2 viral loads across sample replicates was often larger at lower concentrations, sometimes extending over an order of magnitude (Figure 1). Inconsistencies seen here across biological and technical replicates have previously been reported when quantifying SARS-CoV-2 titers in wastewater [23,42,43]. Throughout the study period, the median N2 load was $1.29\times 10^{9}$ gene copies per day and the median N1 daily load came in slightly lower at $1.04\times 10^{9}$ gene copies as quantified through Protocol 1 (Table 1). A general decrease in the day-to-day viral load from February to May is apparent throughout the semester (Figure 1).

### 3.2. Comparison of SARS-CoV-2 Detection

A statistically significant correlation between mean N1 and N2 wastewater concentrations was observed throughout the study period (r = 0.96), and comparable longitudinal wastewater trends are apparent (Figure 1). These data suggest that either gene may be utilized to quantify SARS-CoV-2 to develop trends when using tampons. Additionally, a strong positive association was noted between independent N2 wastewater concentrations on identical samples processed through distinct concentration, extraction, and RT-qPCR methods (r = 0.87) (Figure 2). The average raw N2 wastewater concentration quantified in our lab with Protocol 1 was 3873 ± 10,330 N2 gene copies/mL of wastewater. In comparison, Protocol 2 yielded an average of 3661 ± 6883 N2 gene copies/mL of wastewater. A Wilcoxon rank-sum test found no statistically significant difference between the average N2 concentrations quantified through these separate methods (*p* = 0.17). A high degree of reproducibility in SARS-CoV-2 quantification was evident between the two laboratories suggesting that a standardized method for processing passive wastewater samples may not be overly critical for obtaining valuable data to support public health decision-making. Similar conclusions were noted when 32 independent laboratories processed replicate wastewater samples from two major wastewater treatment plants in Los Angeles County to quantify SARS-CoV-2 using independent methods [43]. After correcting for recovery rates, 80% of the SARS-CoV-2 wastewater concentration data fell within a range of approximately ±1 log GC/L coming from groups utilizing eight distinct methods [43].

### 3.3. Variability in SARS-CoV-2 Signal

Variation in daily N1 and N2 viral loads per individual in isolation throughout the study duration extended greater than four orders of magnitude (Figure 3). The median N2 load throughout the study was measured at $1.01\times 10^{8}$ gene copies per individual per day while the median N1 load per individual was $6.87\times 10^{7}$ gene copies per day (Table 1). The observed variation in fecal shedding is consistent with several studies reporting levels of SARS-CoV-2 in stool ranging from $5.5\times 10^{2}-1\times 10^{7}$ gene copies/mL, which translates to $5.3\times 10^{7}-9.61\times 10^{11}$ expected daily gene copies per infected individual in isolation when accounting for the mean daily building-level water use per person (96 L) at the isolation building [13,14,15,44]. Since we could not obtain data on the temporal complexities of disease progression for each infected individual, fecal shedding rates were assumed constant for the above calculation. Differences in building-level water use indicate significant variation in personal hygiene habits and bathroom behaviors between students (Figure 4). Therefore, substantial variations in individual viral loads per day were expected in this study due to discrepancies in viral shedding, temporal disease dynamics, and water use behavior.

### 3.4. Relationships between Passive Samplers and Epidemiological Reporting

It has been demonstrated that the quantification of the SARS-CoV-2 virus in wastewater is a useful epidemiological tool to develop and complement longitudinal infection trends when using active sampling approaches [42,45]. However, questions remain on whether passive samples are limited to providing binary positive/negative SARS-CoV-2 wastewater results, or if there is utility in quantification and correlation to clinical testing metrics over time. The SARS-CoV-2 wastewater load in the isolation residence peaked on 8 February 2021, at $4.53\times 10^{11}$ N1 gene copies/day, which was coincident with the isolation residence occupancy peak of 222 students (Figure 5). We evaluated the association between the building occupancy and the 24 h SARS-CoV-2 load using negative binomial models, adjusting for the mean BRSV recovery (mean ± SD, 14.0% ± 14.1) and the daily percentage of female occupants in the building as these covariates were significantly associated with occupancy (Table 2). The model was evaluated using McFadden’s pseudo R^2^ as a goodness-of-fit measure [46]; (Appendix A).

A one unit increase in the log-transformed wastewater SARS-CoV-2 daily load was associated with a 47% increase in the total building occupancy, adjusting for BRSV recovery and the percentage of females in the building (Incidence Rate Ratio (IRR) = 1.47 (95% CI: 1.27–1.71, *p* < 0.001). After adjustment for BRSV recovery and the daily SARS-CoV-2 load, we found a 5% increase in occupancy count for each one percentage increase in females occupying the isolation residence (IRR = 1.05; 95% CI: 1.04–1.06, *p* < 0.001). Several factors may have contributed to this observation, including but not limited to differences in fecal shedding intensity between females and males as a result of disease severity or time course of illness. A meta-analysis on SARS-CoV-2 RNA fecal shedding revealed that patients with gastrointestinal (GI) symptoms had a 2.4-fold increased likelihood of excreting detectable levels of SARS-CoV-2 RNA in their stool compared with those with no GI symptoms (Odds Ratio (OR) = 2.4, 95% CI: 1.2–4.7) [47]. Moreover, a cross-sectional study conducted in Poland examining sex differences in COVID-19 symptoms found that self-reported gastrointestinal symptoms among non-hospitalized patients were significantly higher in females than men [48]. Together, these suggest that on average, a greater proportion of females in the isolation residence were shedding detectable levels of virus compared to men as a result of sex-linked differences in the prevalence of GI symptoms from COVID-19. We used our final model to predict the Spring 2021 occupancy in the isolation residence and compared these results with actual case numbers. The results suggest that utilizing tampons for passive sampling in wastewater is a viable option to capture building-level caseloads (Figure 6).

Incorporating additional covariates such as time since symptom onset or diagnosis, sample fecal load, or obtaining more time-resolved flow data could help improve the model by capturing additional changes associated with viral shedding dynamics and SARS-CoV-2 loading onto the passive samplers. Additionally, it is important to note that we captured several detectable SARS-CoV-2 signals in the sewer spanning ten days (11 May 2021–21 May 2021) after the last individual had exited the isolation residence (Figure 1). This observation strongly suggests that viral fragments can remain in sewer systems for an extended period despite no active cases in a catchment. It also suggests that SARS-CoV-2 decay in low-flow environments can occur over a timeframe of weeks. It has been demonstrated that SARS-CoV-2 RNA can accumulate in sewer biofilms which could have served as the primary source of viral RNA during this timeframe [49]. SARS-CoV-2 persistence in the sewer may have important implications for signal interpretation as positive cases move from their regular living quarters to isolation on college campuses or as students return to college campuses after break periods.

### 3.5. Limitations

Several characteristics relating to this research need to be further explored, and various limitations should be considered in generalizing the findings outlined above. First, most passive samples (nearly 90%) remained in the sewer for 24 h; however, some were left at the isolation residence for more than one day. A sensitivity analysis excluding these samples left in the sewer for an extended period demonstrates a 44% increase (47% increase in original model) in the occupancy count for each one unit increase in the log-transformed wastewater SARS-CoV-2 daily load, adjusting for BRSV recovery and the percentage of females in the building (IRR 1.44, 95% CI 1.22–1.70). Though only students with a positive COVID-19 clinical test resided in the isolation building, we cannot exclude the possibility of staff members contributing to the building-level water use, which could bias results. Additionally, it is necessary to note that the quantification of SARS-CoV-2 gene copies/L of wastewater, as measured using the raw influent sewage captured by our passive samplers, may not precisely represent the actual composition of sewage throughout the 24 h sampling period. Instead, our quantification of SARS-CoV-2 gene copies/day in the wastewater comes from the “extracted” wastewater over the 24 h time span, which was required to normalize the N1 and N2 signals to daily flow and report daily viral loads. Moreover, hourly flow data and bathroom-level flush counts may have provided more information on day-to-day student behavior. All samples were frozen at −80 °C for differing periods of time prior to quantification which could have resulted in differential SARS-CoV-2 signal degradation. However, BRSV recovery quantification may account for any degradation in SARS-CoV-2. Lastly, incorporating normalization parameters to account for changes in the daily wastewater contributions could be important to include in future models (e.g., human fecal normalization marker). Still, it remains unclear if the addition of other covariates could better elucidate complex viral shedding dynamics to more accurately estimate building-level caseloads. Even so, data presented here indicate that the duration and direction of trend classification is a practicable surveillance application using a passive sampling approach. Our data indicate a clear relationship between daily viral wastewater concentrations and building-level infection prevalence. We provide a proof of concept that increased SARS-CoV-2 concentrations in sewage, indicated by a greater number of infected individuals, yield an increased accumulation of viral fragments on our passive samplers over 24 h (Figure 4).

## 4. Conclusions

This study investigated the functionality of tampons as passive samplers in capturing SARS-CoV-2 viral fragments in building-level raw sewage. Our method consistently captured virus presence at a COVID-19 isolation residence over 16 weeks, indicating that this method is a feasible option to identify residential halls with infected individuals shedding SARS-CoV-2. The data here also shed light on the quantitative potential of the captured daily wastewater viral load and its relation to building-level COVID-19 prevalence. The positive association found between the daily viral wastewater load and the isolation building occupancy demonstrates the passive samplers’ ability to capture increased SARS-CoV-2 concentrations in influent sewage. Again, additional normalization parameters, including human fecal indicators, could be explored to illustrate building-level COVID-19 caseloads from the captured SARS-CoV-2 wastewater signal. Furthermore, the considerable variability observed in individual viral fecal-shedding loads should be considered in future work if accurately predicting the prevalence of COVID-19 in a residential building is a primary goal.

We demonstrated that SARS-CoV-2 quantification was highly correlated between the N1 and N2 gene and N2 wastewater concentrations measured through different processing pipelines. Here, we provide evidence that universities can use either gene to develop and complement existing COVID-19 trends. We demonstrate that independent wastewater processing and quantification methods provide statistically similar and equally useful wastewater data. The prevailing method for obtaining wastewater samples in WBE involves autosamplers collecting liquid composite samples. However, passive samplers not only ease the burden of deployment efforts and sampling expenses, but such samplers may capture variabilities in the wastewater stream missed by time-weighted autosamplers through continuous exposure to raw influent sewage. Given the sensitivity, low cost, and practicality of deploying tampons as passive samplers coupled with the significant positive correlation observed between the daily wastewater viral load and the caseload in the isolation residence over time, we consider this method to be a valuable public health tool for COVID-19 WBE. In conclusion, this paper provides evidence that tampons can provide reproducible and informative SARS-CoV-2 signs from wastewater, which is particularly relevant for resource-limited communities interested in conducting and operationalizing building-level COVID-19 wastewater surveillance.

## Figures and Tables

**Figure 1 ijerph-19-11245-f001:**
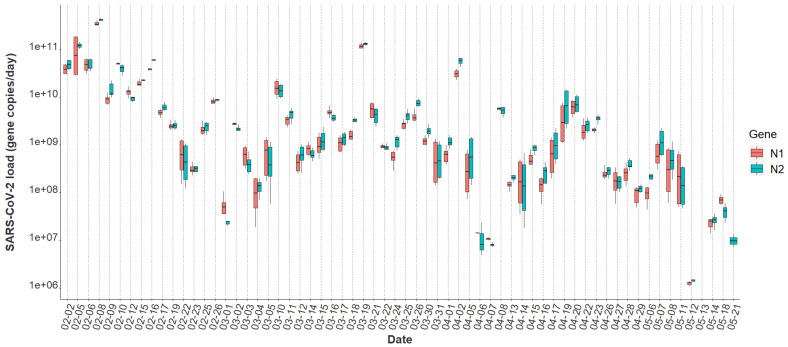
N1 and N2 daily viral loads from the COVID-19 isolation residence, 1 February 2021–21 May 2021 (n = 64).

**Figure 2 ijerph-19-11245-f002:**
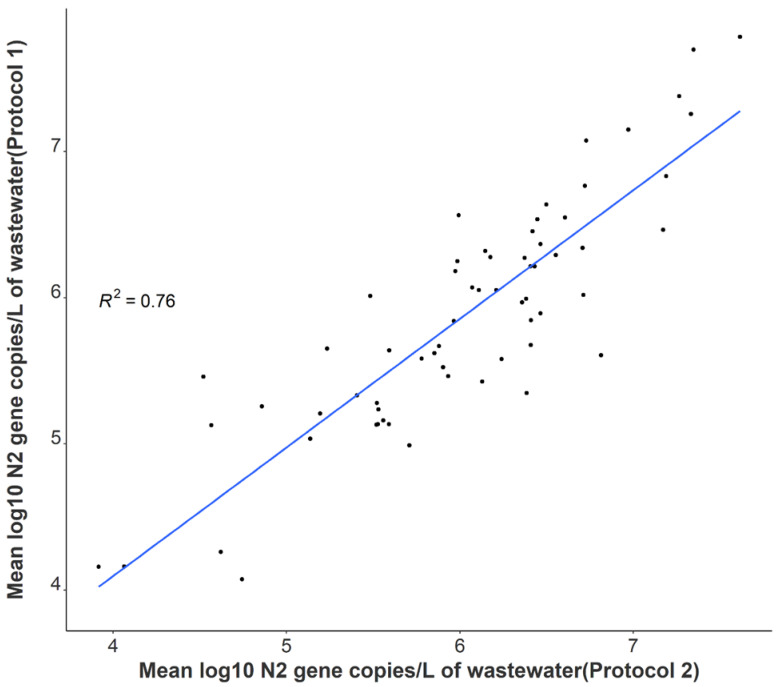
Correlation between independent average N2 log-transformed viral concentrations on identical samples utilizing separate processing and analysis pipelines (n = 64).

**Figure 3 ijerph-19-11245-f003:**
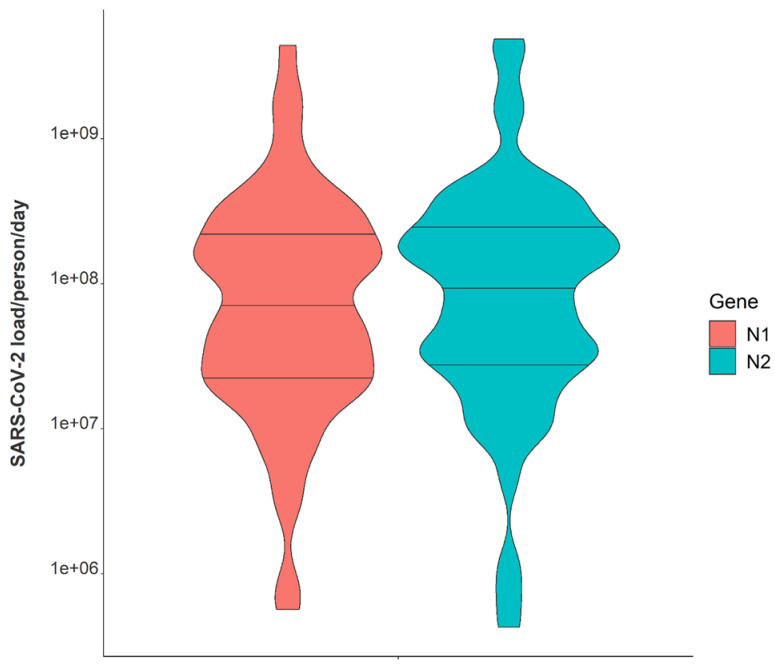
Violin plot showing distributions of N1 and N2 average daily wastewater viral loads per individual in isolation from 1 February 2021–21 May 2021. Note: Markers shown are median, 25th and 75th quantiles.

**Figure 4 ijerph-19-11245-f004:**
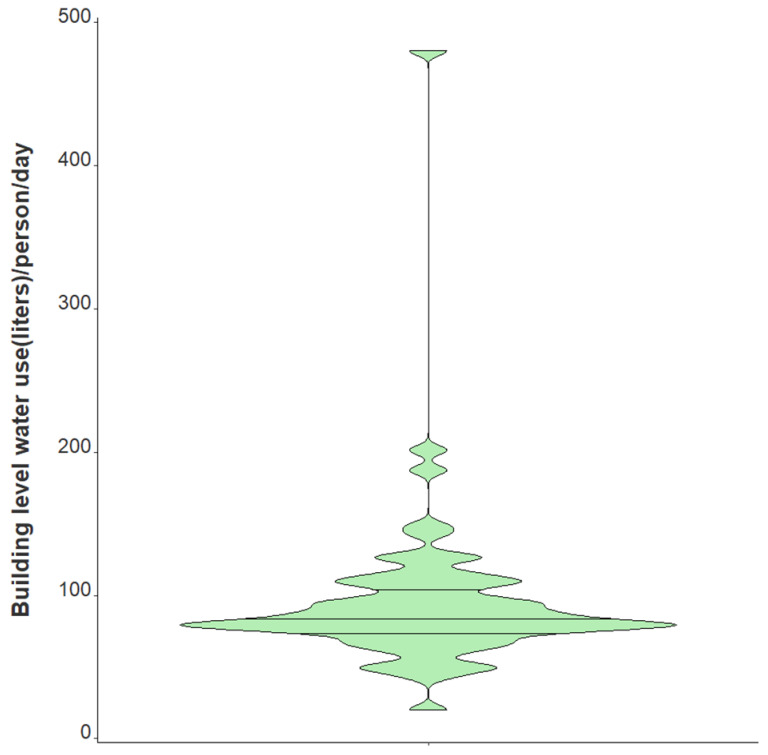
Violin plot showing distribution of average daily building-level water use per individual in isolation from 1 February 2021–21 May 2021. Note: Markers shown are median, 25th and 75th quantiles.

**Figure 5 ijerph-19-11245-f005:**
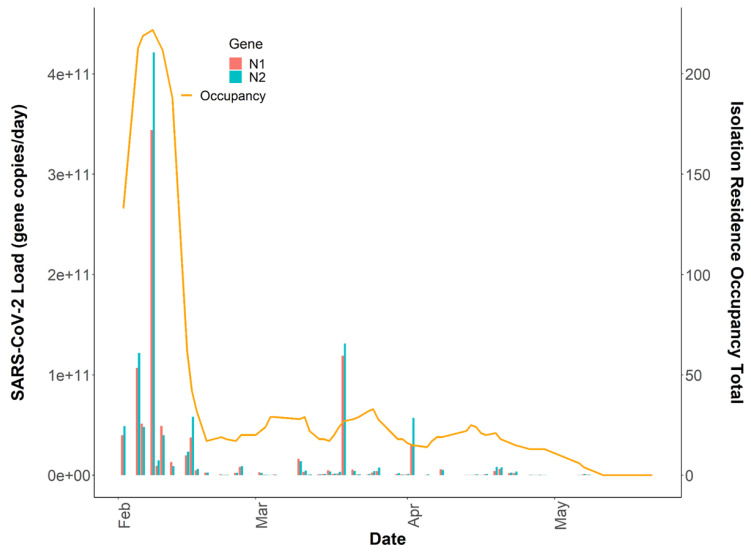
Total daily isolation building occupancy (line) plotted with N1 and N2 gene copies/day (bars) during the Spring 2021 academic semester. Note: Both SARS-CoV-2 daily wastewater viral loads and daily isolation residence occupancy totals are reported on linear y-axes (n = 64).

**Figure 6 ijerph-19-11245-f006:**
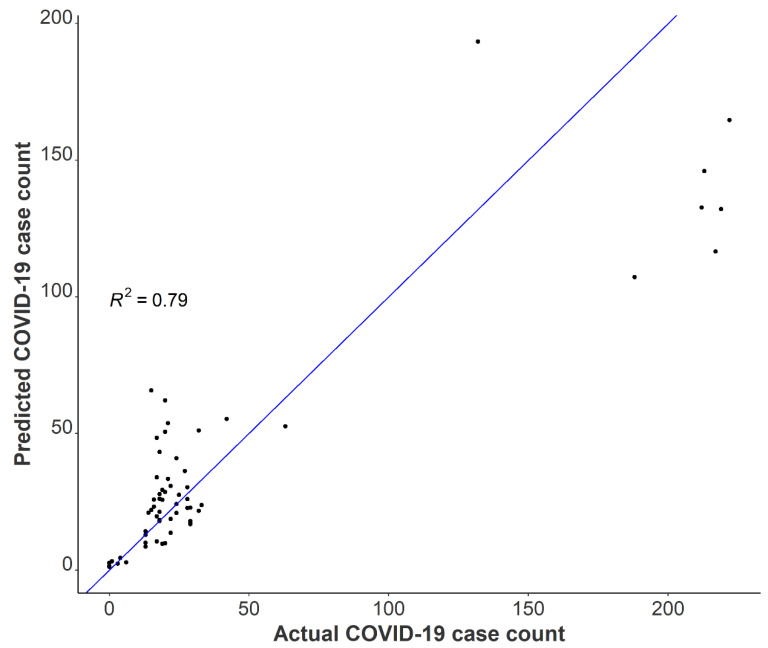
Observed COVID-19 isolation residence occupancy plotted against predicted COVID-19 isolation residence occupancy using negative binomial modeling with Spring 2021 data.

**Table 1 ijerph-19-11245-t001:** Quantification of viral loads and viral loads per person in the isolation residence.

Characteristic	Value
Viral gene copies per day	
Median, N2	1.29 × 10^9^
Range (min/max), N2	1.38 × 10^6^, 4.53 × 10^11^
Median, N1	1.04 × 10^9^
Range (min/max), N1	1.11 × 10^6^, 4.27 × 10^11^
Viral gene copies per person per day	
Median, N2	1.01 × 10^8^
Range (min/max), N2	4.29 × 10^5^, 4.86 × 10^9^
Median, N1	6.87 × 10^7^
Range (min/max), N1	5.70 × 10^5^, 4.41 × 10^9^
Water usage per person per day (L)	
Median	83.4
Range (min/max)	20.3, 479.7

**Table 2 ijerph-19-11245-t002:** Time series negative binomial models to quantify associations between captured viral loads and defined patient populations, Massachusetts 2021. Note: IRR = Incidence Rate Ratio.

Factor	Univariate IRR	Univariate *p*-Value	Adjusted IRR	Adjusted *p*-Value
BRSV Recovery	0.99	0.40	0.99	0.02
% Female Occupants	1.05	<0.001	1.05	<0.001
Log Viral Gene Copies	2.08	<0.001	1.47	<0.001

## Data Availability

The data presented in this study are available on request from the corresponding author. The data are not publicly available due to university restrictions.

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
