# Peer review of "Quantifying the Relationship between SARS-CoV-2 Wastewater Concentrations and Building-Level COVID-19 Prevalence at an Isolation Residence: A Passive Sampling Approach"

_ijerph, 2022, doi:10.3390/ijerph191811245_

Round 1
Reviewer 1 Report
In this manuscript, Acer et al have demonstrated an economical way to perform passive sampling approach for detecting SARS-CoV-2 genome in wastewater sewage. By using two different laboratory pipelines to process the collected samples, the RT-qPCR results from two different sets of probes targeting viral N gene have showed similar outcomes with high sensitivity, which were also corelated with the amount of COVID-19-positive residency at that period.
The manuscript was well-written with clear goals and detailed experimental designs. However, I believe that the way authors analyzing data and interpreted the outcomes could be more comprehensive, especially when the samples were collected from wastewater system, the environmental factors are as important as the residency in the building to affect RNA copy number detection. For example, the sample collecting period was from February to May, is it possible to combine the weather/temperature condition to explain the overall reduction of RNA copy number in that semester? Maybe the temperature rising from late winter to early summer will cause the differences of RNA stability. Another concern is the location to set up wastewater collection, was it in-door or out-door? Will it be affected by rainy weather? If so, does the water flow in the sewage associates with the rain water, or just with water from human activity?
Authors used two different sets of probes to perform RT-qPCR, both are targeting viral N genes at different regions and provided similar outcome of viral RNA copy numbers, which provides a good confidence and consistency. It can be interesting, if possible, to use probes targeting different regions outside of the N gene. Are there variant-specific probes in hand to differentiate the distribution of different SARS-CoV-2 viruses? Especially when the sample collecting period in this manuscript is the time alpha variant taking over ancestor SARS-CoV-2 virus and become dominant in the US, maybe the additional result can represent the transition as well.
One other question. Did authors also perform the sample collection in different ways as they mentioned in the introduction, such as grab samples? Or, did authors tried different absorbent materials other than tampons? By comparing samples collecting from different methods then processing in the same way for RNA detection, one can easily tell the sensitivity and specificity, or pros and cons, in different approaches.
Finally, did authors also collect samples from other wastewater system that is not associating with the quarantine dormitory? Readers may wonder if there are environmental back ground and/or it could also be a good negative control to tell if the isolation building is truly “clean” from virus shedding.
Author Response
We would like to thank you for your insightful remarks and efforts towards advancing this manuscript.
It is true that environmental factors such as ambient temperature or flow impact the composition of 24-hour passive samples; however, the impact appears to be much less at this sampling location directly outside of an isolation residence compared with wastewater treatment plants. As noted by, Schussman and McLellan (Schussman & McLellan, 2022), case-adjusted viral loads can decrease by ~50% in warmer temperatures at wastewater treatment plants, but such a decrease cannot account for the up to 5 orders of magnitude difference observed in viral gene copies per day at our location. Infiltration was not observed to be an issue at this outdoor location (covered manhole) and passive samplers visually appeared consistent throughout the study duration. Two limitations of this paper include no incorporation of a fecal normalization biomarker to account for any potential dilution beyond building-level water use contribution and no measure of the temperature of the sample directly during collection.
It would be interesting to target different regions of the virus and analyze how well that data aligns with the N gene results; however, we do not have sufficient reserves of samples from this study for additional laboratory work and this suggestion seems beyond the scope of this particular study. Regarding differing sample collection methods, we have experimented with different types of tampons and seen results that correlate well. We decided to use the OB Ultra tampons because they provide the largest volume of wastewater. At the time of this study, we were collecting samples from 45 independent locations on campus (one of which was the isolation residence). Beyond the isolation residence, the data we collected at other dormitories using this method correlated well with COVID-19 caseloads further supporting the utility of passive sampling. A different manuscript in preparation by our group supports in detail the function and ability of tampons as effective sample tools. However, we wanted to focus solely on the isolation residence here and our campus-wide data is being incorporated into a separate manuscript. Finally, there is no indication of an environmental background signal at our sampling location as this manhole site only had the isolation residence feeding into it.
Reference
Schussman MK, McLellan SL, 2022. Effect of Time and Temperature on SARS-CoV-2 in Municipal Wastewater Conveyance Systems. Water, 14(9):1373. https://doi.org/10.3390/w14091373.
Reviewer 2 Report
In this manuscript, Acer et al., measured viral loads detected in the wastewater from building. They showed that the viral loads were positively correlated with the number of COVID-19 cases. The data looks clear and experiments conducted well. I have few minor concern/comments for the publication.
Minor
1. To calculate the virus loads (N1/N2) by RT-qPCR, did the authors normalize virus load to internal control (GAPDH etc...)? This information should describe in the Protocol.
2. Figure 1: On 5/13, there are no virus detection. Do the authors have any explanation?
3. Do the author have any information which variants individuals (building occupants) are infected with? Based on the sampling date, they might be infected with Alpha variant. This information (speculation) might be important to describe in the Protocol.
4. Do the authors have any information how many individuals showed severe and mild symptoms?
5. Figure 6: Please add the result of statics analysis (R2).
6. Line330: Please describe non-abbreviate word for “SOPs”.
Author Response
We would like to thank you for your insightful remarks and efforts towards advancing this manuscript.
- We did not normalize the viral load to an internal control. As mentioned, a limitation of this paper was not incorporating a fecal content normalization biomarker into our RT-qPCR assay to adjust for human input; rather we used flow as a proxy for human contribution. Additionally, at this scale, building level water use is a good indicator of human wastewater contribution.
- We are reporting total gene copies/day in Figure 1 and although there was a quantifiable signal on 5/13/21, there was no flow on this day resulting in the normalized signal being 0 as well (raw SARS-CoV-2 signal * flow).
- We do not have any variant-specific data during this timeframe on campus.
- The questionnaire that students in the isolation residence filled out had binary questions such as “Are you experiencing illness-related symptoms” but had no questions pertaining to severity.
- R2value (0.79) had been added to Figure 6
- Abbreviated SOPs changed to methods for brevity.
Reviewer 3 Report
Quantifying the relationship between SARS-CoV-2 wastewater
concentrations and building-level COVID-19 prevalence at an
isolation residence using a passive sampling approach
The present manuscript is about the wastewater-based epidemiology (WBE) approaches to evaluate the relationship between passive SARS-CoV-2 samples isolated and epidemiological metrics. The study was performed using a total of 64 samples collected passively by deploying tampons outside of a COVID- 2019 isolation unit at a university campus between February 1, 2021 – May 2121, 2021. For this author has used RT-qPCR targeting the N1 and N2 gene fragments of SARS-CoV-2 to monitor the mean viral load captured per individual and the association between the daily viral load using multivariable models to provide a baseline estimate of viral shedding. The study finds a positive association of daily SARS-CoV-2 RNA loads in building-level wastewater with the total number of COVID-19-positive individuals in the residence
The present article is interesting to me, I found the article title lengthy, please consider rewriting the title appropriately. This manuscript is easy to follow. Following are the specific comments to further strengthen the present manuscript,
1. The primary question is, does the disease course affect these results? It is evident that COVID-19 patients shed the virus in stool, but how the viral load determines the RNA in passively collected samples? Can the correlation between WBE and infected individual calculation be wrong considering higher/lower viral shedding in different stages of infected individuals?
2. Does the author notice any difference between the viral load from 52ml and 80ml of tampons samples? There may be some variations in this volume from some tampon samples, how was this considered during the study?
3. Any explanation for the higher sex-linked proportion of viral shedding observed in females? Is this due to the higher prevalence of GI symptoms in COVI-19 infected females?
4. The author can discuss the present study's use for asymptomatic COVID-19 patients.
Author Response
- We agree that individual disease time course could play a significant role in the viral load we captured through our passive samplers and potentially bias the results. Our inability to track the progression of infection for each individual in isolation is a primary limitation of this paper. However, we do not believe our correlation between WBE data and the infected individual calculation to be "wrong", rather perhaps simplified, as indicated by an R2 of 0.79 in Figure 6. We acknowledge that there is unmeasured variability in this dataset yet Figure 6 indicates that our model still provides similar building-level COVID-19 data compared with observed data in the residence. We would expect improved modeling capability with disease time course data; however, our computation would still not be perfectly accurate.
- This is an interesting point. Our instincts would guide us to believe there would not be a difference in viral load based on initial sample volume since both viral concentration/capture protocols used a fixed volume of wastewater independent of the originally sample volume. We plotted the total sample volume against its corresponding raw SARS-CoV-2 wastewater signal (attached below). We do not see compelling evidence that initial sample volume impacted the recovered viral load.
- In lines 397-400 we indicate that on average females were shedding more virus in the building compared to males due to sex-linked differences in the prevalence of GI symptoms.
- We have found that there is already an abundant amount of WBE literature outlining the benefits of wastewater surveillance in capturing asymptomatic individuals. The scope of this paper is to focus on the quantitative potential of passive sampling in capturing building-level infection dynamics accurately.
